# Investigating epigenetic biomarkers of age, sex, and disease in captive South African cheetahs (*Acinonyx jubatus jubatus*)

**Michelle Cristi Ysrael**[ORCID]°, **Steven Kubiski**°, **Caroline E. Moore**[ORCID]°*

San Diego Zoo Wildlife Alliance, Escondido, California, United States of America

° These authors contributed equally to this work.
* camoore@sdzwa.org

## Abstract

Epigenetic modifications, particularly DNA methylation, are strongly associated with chronological age across mammalian species. This study developed cheetah-specific epigenetic clocks from methylation profiles generated from cheetah blood and liver samples tested on the HorvathMammalMethylChip40 Illumina Array. The resulting age clock used 52 CpG sites and predicted age across blood and liver samples ($r = 0.97$ and MAE = 0.86). When applied to a test set of blood collected from live cheetahs, the clock provided accurate predictions for adult individuals (age > 3 years) but was less precise at and around age of sexual maturity. A second clock, incorporating cheetah, lion, and tiger profiles, used 46 CpG sites and predicted age across these feline species ($r = 0.94$ and MAE = 1.16). Additionally, a sex clock using 67 CpG sites accurately predicted sex in all test samples. To explore the potential of methylation as a biomarker beyond age and sex, we conducted a differential methylation analysis to investigate disease-related methylation patterns in cheetahs diagnosed with hepatic sinusoidal obstruction syndrome (SOS). This analysis identified 4,377 CpG sites with significant differences between SOS-positive and SOS-negative cheetahs (adjusted p-value < 0.05). These findings advance the development of epigenetic clocks for precise age and sex prediction in cheetahs and related species and establish a foundation for leveraging methylation biomarkers to investigate diseases in wildlife conservation efforts.

## Background

Epigenetic modifications are defined as molecular modifications related to gene activity that do not involve changes in DNA sequence but can play a significant role in gene regulation. These modifications play a dual role in biological processes, aiding in adaptive responses to environmental changes while also contributing to non-adaptive processes like aging [1,2]. Further, dysregulation of these mechanisms has

**Data availability statement:** "Methylation data files, including raw IDAT, SESAME normalized files and sample metadata sheet, are available on NCBI GEO (accession number: GSE310779).".

**Funding:** This project was funded by a generous philanthropic grant from the Elwyn Heller Foundation of San Diego for the Bud Heller Conservation Fellowship awarded to Michelle Ysrael (MY) (BHF36002). No website URL. The funders did not play a role in study design, data collection/analysis, or preparation of the manuscript for publication.

**Competing interests:** The authors have declared that no competing interests exist.

been linked to disease, as epigenetic activation or repression of specific genes and pathways can contribute to disease initiation and progression [3,4].

The discovery of age-related epigenetic patterns has led to the development of so-called epigenetic clocks that have applications across human and animal medicine. DNA methylation is one epigenetic modification that demonstrates site-specific linear changes that correlate with age [5–7]. DNA methylation involves the covalent binding of methyl groups to cytosine residues. These regions are termed "CpG sites", with the "p" denoting the phosphate linker between the two residues. The addition of methyl-groups results in structural changes to DNA that can affect gene expression. For example, methylation at transcription factor binding sites can block transcription initiation [8].

Epigenetic clocks have been developed for age prediction in various species by utilizing methylation patterns at specific age-related CpG sites [9–13]. Pan-species (i.e., applicable across multiple species) and pan-tissue (i.e., applicable across multiple tissue types) clocks such as the universal pan-mammalian epigenetic clocks developed by Lu et al [13] provide versatile tools for cross-species comparisons. Arrays used to build these clocks, including the HorvathMammalMethylChip40 developed by Arneson et al, were specifically designed for cross-mammalian application using conserved CpG sites selected through multi-species alignments [14]. Species-specific epigenetic clocks (made from HorvathMammalMethylChip40 data) have been developed to increase applicability of these tools in wildlife research and medicine, allowing for more precise age estimates using fewer, more targeted CpG sites [15]. Raj et al. developed a feline epigenetic clock for domestic cats but reported limitations and reduced accuracy when applied to cheetah, lion and tiger samples, highlighting the need for species-specific clocks for more accurate age prediction [11].

Accurate determination of chronological age is a critical component of wildlife management and conservation as it facilitates the assessment of health, reproductive status, and large-scale population dynamics [16]. Currently, skull measurements, radiographs, and dental examinations are the most common vertebrate age estimation methods [17–20]-- however, these methods are often difficult to apply accurately and consistently at scale in field settings. Epigenetic age clock measurements offer an alternative to these methods, with potential applications for determining not only chronological age but also health status.

Building epigenetic clocks for wildlife is challenging due to limited biobanked samples, especially blood. At the San Diego Zoo Wildlife Alliance, we leveraged liver samples from our biobank and cheetah blood profiles from the Mammalian Consortium Data Browser (MCDB), all from captive individuals with known chronological age and sex, to construct an epigenetic clock applicable to less-invasive samples like blood and skin (compared to liver). While prior studies suggest 134 samples (minimal 70) are needed for accurate age clocks [21], we tested the reliability of a smaller dataset (n = 52) to address common limitations in wildlife research. We also explored combining tissue types and feline species to expand age prediction capabilities, enhancing monitoring options for cheetahs and other wildlife using biobanked

resources. Additionally, we used our extensive clinical and pathology database to investigate possible health status bio-markers in San Diego Zoo and Safari Park-housed cheetahs. In summary, this work aims to improve methods for epigenetic age estimation in wildlife with limited samples to support wildlife conservation research and management.

The goals of this study were to utilize epigenetics in two ways: 1) investigate DNA methylation patterns to develop cheetah-specific epigenetic clocks for age and sex prediction and expand this across clinically and field relevant tissues and wild felid species; and 2) to explore the utility of methylation analysis to inform disease investigations and diagnostics in zoo-housed cheetahs. For this second goal, we focus on sinusoidal obstructive syndrome (SOS), a recognized disease of zoo-housed cheetahs that can be subtle or subclinical in early stages [22]. We hypothesized that DNA methylation-based models could accurately predict age and sex in cheetahs, even with a limited sample size, and that incorporating multiple tissue types and related felid species would enhance predictive power and generalizability. Additionally, DNA methylation has been studied for its utility as a biomarker in various human and animal diseases [23,24], and we hypothesized that epigenetic differences would distinguish SOS-affected and unaffected cheetahs. Findings from this study present avenues for developing more reliable biomarkers of age, sex, and general health, as well as offer a foundation for continued research into SOS etiology.

## Methods

### Study population and samples

This animal study was reviewed and approved by the San Diego Zoo Wildlife Alliances biomaterials request. Samples used in this study were a part of a biomaterials request under an approved protocol for opportunistic sampling during veterinary and husbandry procedures (IACUC #24–103). The study population consisted of 44 cheetahs that had biobanked tissue samples available for use. All animals were born and held under human care until death. Liver tissues were collected opportunistically from 42 cheetahs that died between 1993 and 2024 at the San Diego Zoo or San Diego Zoo Safari Park, collectively San Diego Zoo Wildlife Alliance (SDZWA). Samples were collected as part of a complete necropsy performed routinely on all animals that died at our facilities. For five of these cheetahs, skin samples also collected during necropsy were included.

Biobanked whole blood samples (n = 7) collected from seven live cheetahs opportunistically during exams were also included. Of these seven blood samples, five were collected while the cheetahs were alive and later correlated with necropsied liver samples, while two were from cheetahs currently alive, with no corresponding liver samples. Table 1 summarizes the tissue types used in the study, and depicts repetitive sampling from the same individuals (Table 1). Ages of the cheetahs ranged from 0 days old to 16 years (mean: 9 years, interquartile range (IQR): 8 years), and included 26 females and 18 males (S1 Table). Full sample descriptions can be found in S1 Table.

All tissue samples were stored at −20°C or −80°C. Genomic DNA was extracted using Monarch Genomic DNA Purification Kit (New England Biolabs, Ipswich, MA; Cat.T3010S) or DNeasy Blood and Tissue Kit (Qiagen, Valencia, CA; Cat.

**Table 1. Summary of tissues tested for individual cheetahs.**

| Cheetah ID | 1 | 2 | 3 | 4 | 5 | 6 | 7 | 8 | 9 | 10 | 11 | 12-42 | Count |
|---|---|---|---|---|---|---|---|---|---|---|---|---|---|
| Liver samples | ✓ | ✓ | ✓ | ✓ | ✓ | ✓ | ✓ | ✓ | ✓ | | | ✓ | 42 |
| Skin samples | ✓ | ✓ | ✓ | ✓ | ✓ | | | | | | | | 5 |
| Blood samples | | | | | ✓ | ✓ | ✓ | ✓ | ✓ | ✓ | ✓ | | 7 |

Summary of tissue types used in the study for individual cheetahs (n = 44). Checkmarks denote the tissue types collected for each cheetah (labeled 1–44): Cheetahs 1–4 had both liver and skin samples tested, cheetah 5 had liver, skin and blood samples tested, cheetahs 6–9 had both liver and blood samples tested, cheetahs 10–11 only had blood samples tested, and cheetahs 12–44 had only liver samples tested. The rightmost column is the total count for each tissue type.

No./ ID: 69504) with the following modifications for the DNeasy kit: 40 uL Proteinase K was used instead of the 20 uL suggested in the manual.

Medical records were reviewed for the 44 cheetahs to identify health outcomes for further evaluation. These records documented a variety of health backgrounds: two cheetahs had no underlying disease and the rest were reported to have one or more diseases at time of death, including bacterial infections, gastritis, chronic kidney disease, and sinusoidal obstructive syndrome (SOS). The disease SOS was the focus of the present study here due to the high number of cheetahs with SOS diagnosis (n = 11, 25%). Two of the seven blood samples included were collected from cheetahs that were not diagnosed with SOS while they were alive but were incidentally found to have SOS upon necropsy. All cases were reviewed again by a board-certified veterinary pathologist and cross referenced with original board-certified veterinary pathologist reports.

## Mammalian consortium data browser data

Methylation data publicly available on the Mammalian Consortium Data Browser (MCDB) [25] was also used to supplement our study. This database contains DNA methylation profiles for 348 mammalian species generated using the mammalian methylation array [14]. For this study we downloaded blood methylation profiles available for cheetahs (accessed: July 2023), lions (accessed: August 2023), and tigers (accessed: August 2023).

## The clock foundation DNA methylation testing

DNA was quantified on a Qubit 4 fluorometer (ThermoFisher) and plated on a fully skirted, low profile PCR style plate sealed with adhesive PCR plate foil seal (ThermoFisher). Samples were placed in random order to prevent any bias based on the chip. Approximately 250 ng of DNA was sent to The Clock Foundation (Torrance, CA) on dry ice overnight for processing through their previously published pipeline for DNA methylation testing [14]. Briefly, the Infinium method uses sodium bisulfite conversion and microarray-based single-base genotyping to assess CpG methylation: unmethylated cytosines are deaminated by bisulfite and read as thymines after PCR, and methylated cytosines are protected by conversion and remain cytosines. This platform enables high-throughput, multiplexed analysis using bead-bound probes and fluorescent single-base extension to distinguish cytosine to thymine conversions at target CpG sites [14]. The mammalian methylation array HorvathMammalMethylChip40 (Horvath Array) used to build methylation profiles from our genomic DNA samples was developed by Arneson et al [14]. The array targets 37,492 CpG sites, 35,988 of which are well conserved across many mammalian species [26]. Species-specific probe annotation is provided in the chip manifest file found on Gene Expression Omnibus (GEO) at NCBI at platform GPL28271. The raw data were normalized using the SeSaMe pipeline to produce beta values for each probe [27].

## Clustering-based detection of sample outliers

Unsupervised hierarchical clustering of the methylation profiles was performed separately for each tissue type (liver, skin, and blood) to identify samples with distinct methylation patterns that diverge from the main group due to potential technical artifacts, sample quality issues, or distinct biological differences. This analysis identified five outlier samples—four liver samples and one skin sample—that clustered distinctly from the main groups (S1 Fig). These samples were excluded from further analysis (Table 2).

## Predictive models

All statistical analyses and model building were conducted in R software (R version 4.0.1) [28], and details on the clocks, including R software code, are available on GitHub [29]: https://github.com/mysrael/CheetahClock. We applied elastic net regression models from the R package glmnet [30] to build predictive age clocks from the generated methylation profiles

**Table 2. Sample count following exclusion.**

|  | Count | Clustering outliers | Final total |
| --- | --- | --- | --- |
| Liver samples | 42 | 4 | **38** |
| Skin samples | 5 | 1 | **4** |
| Blood samples | 7 | 0 | **7** |

Summary of final sample counts after unsupervised hierarchical clustering outlier exclusion. Four liver samples and one skin sample displayed distinct clustering and were excluded. Final sample counts for each tissue are summed in the rightmost column.

as used in previous epigenetic clock studies [10–13,31]. Log-linear age transformation was performed to account for "accelerated aging" methylation patterns observed before age of sexual maturity (ASM) [32], as done by Lu et al [33] in building the pan-mammalian epigenetic clocks, and by Peters et al [31] in building bottlenose dolphin epigenetic clocks. ASM was determined from previous studies that found both male and and female cheetahs reached physical and sexual maturity at around 2 years [17]. Accuracy was measured using correlation $r$ (Pearson correlation between inverse transformed predicted age and known age) and the median absolute error (MAE) between predicted and known ages.

## Epigenome-wide association studies (EWAS) of age and sex

EWAS is used to investigate genome-wide epigenetic variants that are statistically associated with phenotypes or traits of interest [34]. EWAS was performed separately on liver (n = 38), blood (n = 7) and skin (n = 4) samples using the R function standardScreeningNumericTrait from the WCGNA [35] R package to find DNA methylation changes that correlate with age using Pearson correlation test. For liver and blood EWAS, p-value $10^{-3}$ was the significance threshold. For the smaller skin EWAS, p-value $10^{-2}$ was the significance threshold. Direction of methylation change (hyper- or hypomethylation) was measured using $z$ direction of association value from the output from the age EWAS, where positive $z$ values indicate increased methylation (hypermethylation) with aging and negative $z$ values indicate decreased methylation (hypomethylation). EWAS analysis was also conducted on sex-related CpGs with standardScreeningBinaryTrait using Student's t-test to compare methylation levels between females and males [35].

## Differential methylation analysis

To investigate differential methylation between SOS-positive and SOS-negative individuals, a differential methylation analysis (DMA) was conducted on liver samples using the lmFit with function from the limma [36] package in R. SOS was diagnosed in individuals from this cohort as early as 5 years old, with suspect liver changes as early as 3 years old. To identify potential epigenetic changes associated with SOS but avoid ASM bias, we limited samples to over 3 years of age (n = 30).

SOS status at necropsy was determined by histopathology reports. Based on board-certified pathologist-reviewed reports, samples were annotated with one of the following descriptions: diagnosis of SOS, SOS suspect, liver changes/diagnosis (non-SOS), or no liver comments/diagnosis. Diagnosis of SOS was defined by fibrosis surrounding or occluding hepatic sinusoids and central and/or sublobular veins. 'SOS suspect' was used to characterize individuals with SOS-like liver changes, typically restricted to minimal or mild changes that could be considered non-specific. This was used to account for the evolving characterization of SOS [37]. For the DMA, only samples with an explicit mention of SOS diagnosis (diagnosis of SOS) were classified as SOS-positive. Additional categories ("SOS suspect", "non-SOS liver changes," and "no liver diagnosis") were annotated only in visualizations.

CpGs that displayed significant methylation patterns (hyper- or hypomethylation) between SOS-positive and SOS-negative groups were selected based on p-adjusted value (p.adj) from Benjamini-Hochberg false discovery rate

correction. The methylation data was subset to include the top 100 significant CpG sites from the DMA to be visualized on a heatmap and clustered using unsupervised hierarchical clustering [38] to observe methylation patterns for these CpG sites across the samples. Methylation data from the seven blood samples were also subset to include the CpG sites from the DMA and added to the heatmap for comparison. Four of the blood samples were correlated with liver samples used in the DMA, collected 5–13 years before the cheetahs' death and necropsy.

### CpG annotations

To estimate genomic coordinates of CpGs on the cheetah genome we used alignment information of the array probes to *Felis_catus*_9.1.100 genome assembly from the array manifest file (MammalianMethylationConsortium GitHub) [39]. Human Hg38 genome annotations from the array manifest were used to annotate CpG sites due to the high conservation of probes on the array across mammalian species. These annotations provide a reference point for functional insights, acknowledging that further refinement of annotations will be necessary as the cheetah genome becomes more extensively annotated.

### Pathway enrichment analysis

Additionally, we conducted a Kyoto Encyclopedia of Genes and Genomes (KEGG) pathway enrichment analysis of genes associated with differentially methylated CpGs to identify biological pathways potentially affected by SOS-related methylation changes. Enrichment analysis was done using the function enrichKEGG from the R program clusterProfiler [40]. Analysis of all significant (p.adj < 0.05) CpG-associated genes, as well as separate analyses on hypermethylated and hypomethylated CpGs, was conducted to discern methylation alterations and their potential implications.

## Results

### UniversalClock and CatClock performance on cheetah samples

When applied to our dataset, the pan-tissue, pan-mammalian epigenetic clocks developed by Lu et al [13] (UniversalClock2 and UniversalClock3) performed well on liver, blood, and skin samples in individuals below the age of sexual maturity for cheetahs (2–3 years); however, in older individuals, predictions were more variable, which contributed to the high median absolute error (MAE) of 3–4.42 years (S2 Fig). The cat epigenetic clock by Raj et al (CatClock) trained on domestic cat blood [11] demonstrated strong performance on blood samples, with a correlation of 0.79 and MAE of 1.38 years (S2 Fig). However, its performance was weaker when applied to the combined dataset of blood, liver, and skin samples, with a lower correlation of 0.64 and a higher MAE of 3.5 years (S2 Fig).

### CheetahClock performance across tissues and related felid species

We built cheetah-specific clocks for age and sex prediction using an elastic net regression model with Leave-One-Out-Cross-Validation (LOOCV). The CheetahClock was trained on the methylation profiles generated from the SDZWA cheetah liver samples (n = 38) and cheetah blood methylation profiles from MCDB (n = 14), and tested on SDZWA cheetah blood (n = 7) and skin (n = 4), as well as MCDB lion (n = 7) and tiger (n = 8) blood methylation profiles (Table 3).

In addition to the literature-determined age of sexual maturity (ASM) of 2 years, the clocks were also tested using ASMs of 1.5 and 3 years. The differences in CpG selection and test results (e.g., *r* and MAE) were minimal, with little effect on the clocks' predictive performance (S3 Fig).

The resulting CheetahClock for age used 52 CpG sites (S2 Table). Cross-validation analysis indicated age correlation Pearson correlation (*r*) between predicted age (DNAm age) and chronological age. Age estimations were accepted as accurate if they were within 20% of true age (80% confidence interval) which resulted in an estimated age within 1 year of true age. The CheetahClock produced accurate age estimations in LOOCV analysis on cheetah SDZWA liver and MCDB blood samples (*r* = 0.97, MAE = 0.86) (Fig 1A) (S3 Table). When applied to a test set of SDZWA cheetah blood samples,

**Table 3. Summary of methylation profiles used to build CheetahClock.**

| Methylation profiles | CheetahClock Training | CheetahClock testing |
|---|---|---|
| SDZWA cheetah liver | 38 | 0 |
| MCDB cheetah blood | 14 | 0 |
| SDZWA cheetah blood | 0 | 7 |
| SDZWA cheetah skin | 0 | 4 |
| MCDB lion blood | 0 | 7 |
| MCDB tiger blood | 0 | 8 |
| Total | 52 | 26 |

Samples used to build and test the CheetahClock. 38 SDZWA cheetah liver and 14 MCDB cheetah blood methylation profiles comprised the training set used to build the clock. The resulting clock (CheetahClock) was then tested on the seven SDWA blood profiles and four SDZWA skin profiles. Additionally, the clock was tested on seven lion and eight tiger blood methylation profiles from MCDB.

age was accurately predicted in five out of seven samples (Fig 1B) (S4 Table). In the other two samples, which are around ASM, age prediction was overestimated by 3.1 (age = 2.5 years) and 5.6 (age = 1.25 years) years.

We also assessed model performance on a test set of four SDZWA skin samples. The age prediction results for these skin samples produced an $r = 1$ and MAE = 1.77 (Fig 1C). To assess applicability of the CheetahClock for age in other large cat species, we used lion and tiger methylation data from the MCDB. Prediction results indicated high correlation, but also high MAE for both lions ($r = 0.99$, MAE = 7.55) (Fig 1D) and tigers ($r = 0.96$, MAE = 6.74) (Fig 1E).

A CheetahClock for sex was trained on the same combined SDZWA liver and MCDB blood data and used 67 CpG sites (male n = 16, female n = 31) (S5 and S6 Tables). Sex was accurately predicted in the seven blood (male n = 5, female n = 2) and four skin (male n = 2, female n = 2) test samples, as well as in all tiger (male n = 5, female n = 4) and lion (male n = 1, female n = 6) samples from MCDB (S7 Table).

### FelidClock performance across felid species

We used the same methods as the CheetahClock to build a predictive age model for cheetahs, lions, and tigers using methylation profiles from MCDB (tiger n = 8, lion n = 7, cheetah = 14) and SDZWA cheetah liver samples (n = 38) (Table 4).

The resulting FelidClock used 46 CpG sites (33 overlapping with CheetahClock CpGs) (S8 Table). In LOOCV analysis, the FelidClock achieved high correlation and low error ($r = 0.94$, MAE = 1.16) (Fig 2A) (S9 Table). The clock produced similar results to the CheetahClock when applied to the cheetah blood test samples ($r = 0.76$, MAE = 1.19) and the cheetah skin test samples ($r = 0.99$, MAE = 1.77) (Fig 2B-C) (S10 Table).

### Age-related CpGs

EWAS of chronological age was performed on liver, blood and skin samples. EWAS on liver (n = 38) found 2827 CpGs exhibited methylation patterns associated with chronological age with significance ($p < 10^{-3}$), with 1440 hypomethylated and 1387 hypermethylated CpGs (Fig 3A). Significant CpGs were distributed across genic and intergenic regions defined relative to transcription start site: genic regions included 3-prime untranslated regions (3'-UTRs), 5'UTRs, promoters, introns, and exons; intergenic regions are located between genes, specified as upstream or downstream relative to gene transcriptional start site. Proportion of hypermethylated CpGs was higher in promoter (79%), 5'UTR regions (77%), and exon regions (52%) (Fig 3B).

EWAS on blood was performed on a combined dataset of our seven blood samples and the 14 cheetah blood samples from MCDB (n = 21) and found 715 significant CpGs ($p < 10^{-3}$), with 395 hypomethylated and 320 hypermethylated

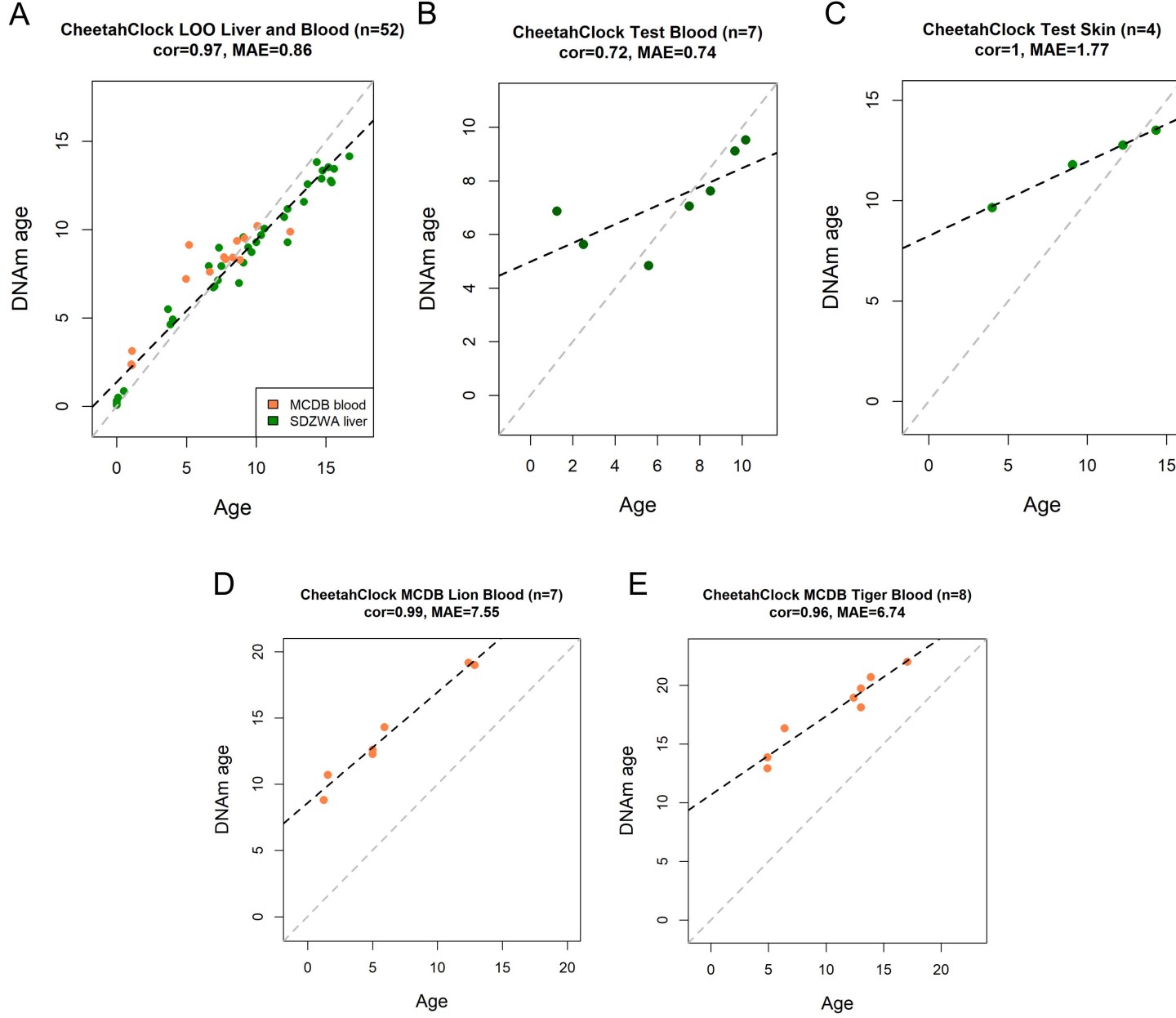

**Fig 1. CheetahClock results.** Evaluation of CheetahClock on methylation profiles. **A)** LOOCV estimation of predicted age (DNAm age) versus chronological age for cheetah blood and liver samples. **B)** CheetahClock applied to SDZWA cheetah blood samples, **C)** SDZWA cheetah skin samples, **D)** MCBD lion blood methylation data, and **E)** MCBD tiger blood methylation data. Each panel reports sample size, correlation coefficient, and median absolute error (MAE).

CpGs (Fig 3C). Proportion of hypermethylated CpGs was higher in promoter (81%), 5'UTR regions (57%), and intergenic upstream (52%) and downstream regions (53%) (Fig 3D).

An exploratory EWAS on the four cheetah skin samples from our study was conducted using a higher significance threshold due to the limited dataset. 369 significant CpGs were significantly associated with age at $p < 10^{-2}$, with 232

**Table 4. Summary of methylation profiles used to build FelidClock.**

| Methylation profiles | FelidClock Training | FelidClock Testing |
|---|---|---|
| SDZWA cheetah liver | 38 | 0 |
| MCDB cheetah blood | 14 | 0 |
| MCDB lion blood | 7 | 0 |
| MCDB tiger blood | 8 | 0 |
| SDZWA cheetah blood | 0 | 7 |
| SDZWA cheetah skin | 0 | 4 |
| Total | 67 | 11 |

Samples used to build and test the FelidClock. 38 SDZWA cheetah liver, 14 MCDB cheetah blood, seven MCDB lion blood, and eight MCDB tiger blood methylation profiles comprised the training set used to build the clock. The resulting FelidClock was then tested on the seven SDWA blood profiles and four SDZWA skin profiles.

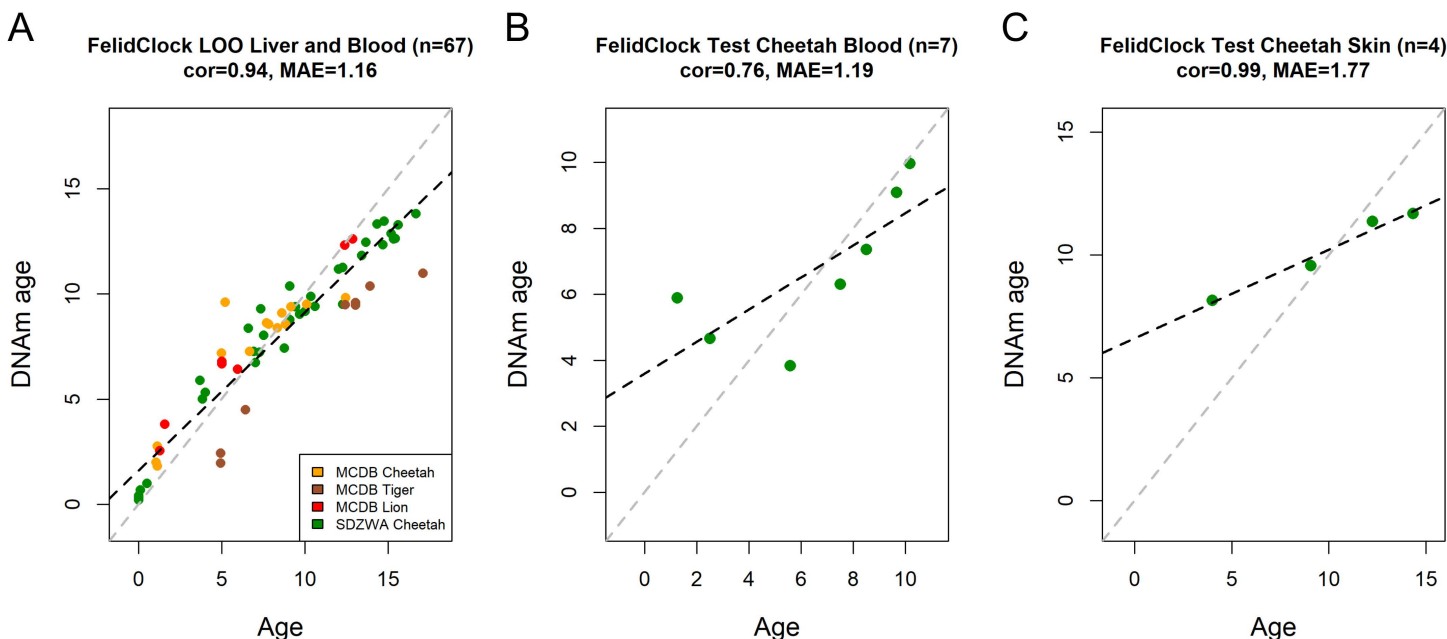

**Fig 2. FelidClock results.** Evaluation of FelidClock on methylation profiles. A) LOOCV estimation of predicted age (DNAm age) versus chronological age in cheetah, lion and tiger samples. B) FelidClock applied to SDZWA cheetah blood samples, and C) SDZWA cheetah skin samples. Each panel reports sample size, correlation coefficient, and median absolute error (MAE).

hypomethylated and 137 hypermethylated CpGs (Fig 3E). Proportion of hypermethylated CpGs was higher in intergenic upstream (51%), with other regions exhibiting higher proportions of hypomethylated CpGs (Fig 3F).

Overlapping CpGs from EWAS in liver, skin and blood were analyzed (Fig 3G). There were 269 CpGs found in both blood and liver EWAS, 79 CpGs in both liver and skin EWAS, and four CpGs in both skin and blood EWAS. The four CpGs found in all three EWAS are associated with the genes *NEUROD1*, *TNR*, *C1D*, and *UNCX*, and displayed the same direction of association across tissues in all three EWAS (Fig 3G).

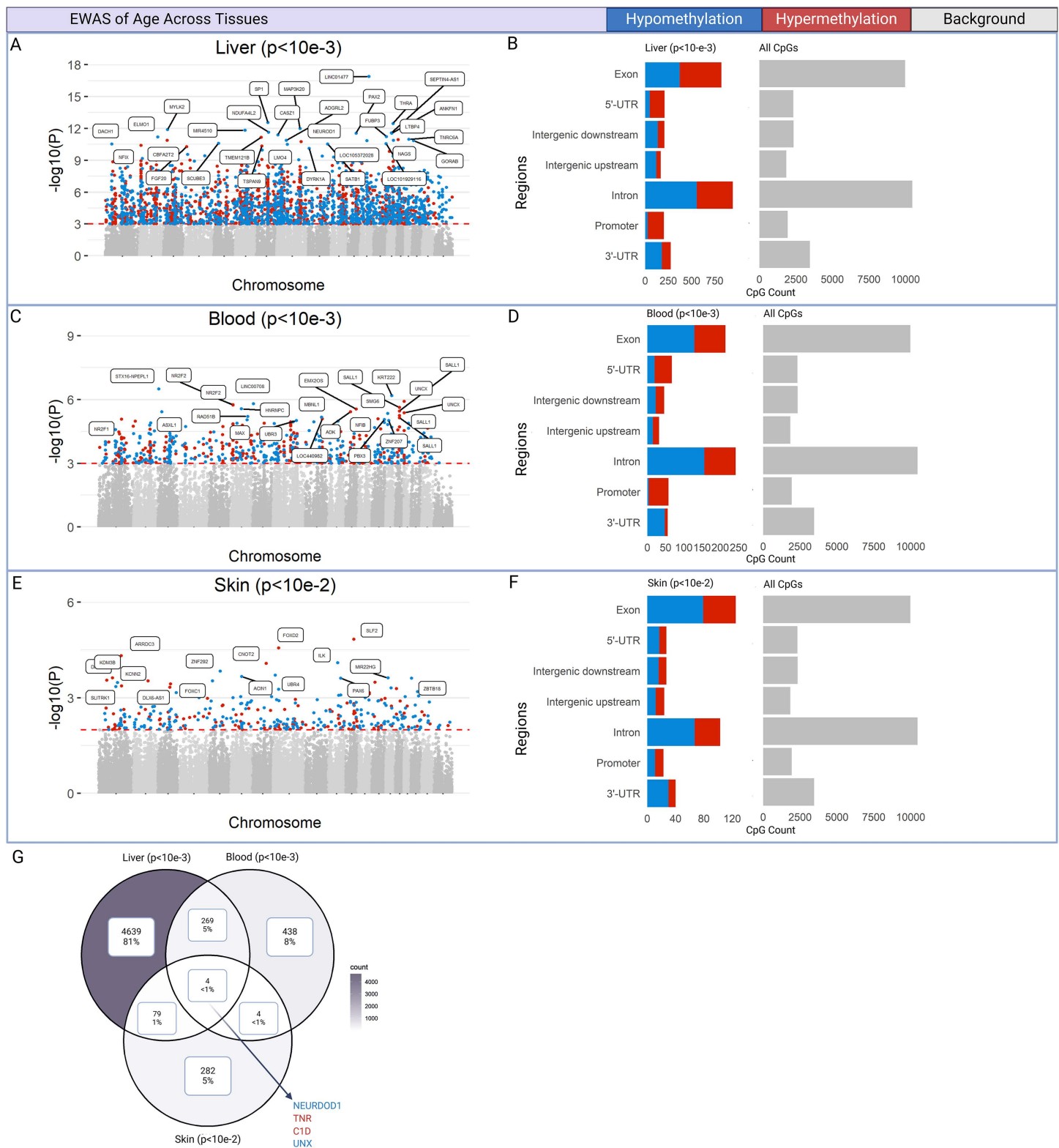

**Fig 3. EWAS analysis.** Epigenome-wide association (EWAS) analysis of chronological age in cheetahs. **A, C, E)** Manhattan plots of the EWAS of age in liver samples (n = 38), blood samples (n = 21), and skin samples (n = 4), with each point representing a CpG site. CpG genomic coordinates are estimated from the alignment of array probes to the feline genome (*Felis_catus_9.1.100*). Significant CpGs—liver (p < 10⁻³), blood (p < 10⁻³), and skin

(p<10<sup>-2</sup>)—are highlighted to depict the direction of association: red indicates hypermethylation (z>0), and blue depicts hypomethylation (z<0). CpGs with the lowest p-values are labeled with their adjacent gene names. **B, D, F)** Location of significant CpGs relative to the transcriptional start site from the EWAS of age in liver ($p < 10^{-3}$), blood ($p < 10^{-3}$), and skin ($p < 10^{-2}$). Red bars represent hypermethylated CpG proportions, blue bars indicate hypomethylated CpG proportions, and gray bars show the distribution of the total 34,851 array probes mapped to the feline genome (*Felis_catus_9.1.100*). **G)** Venn diagram illustrating overlap in significant CpG sites from EWAS of age in liver ($p < 10^{-3}$), blood ($p < 10^{-3}$), and skin ($p < 10^{-2}$). CpGs shared among all three tissues are annotated with associated gene names. In all overlapping results, these CpGs showed consistent directions of association across tissues, indicated by font color (red = hypermethylation, blue = hypomethylation).

### SOS-related CpGs

A differential methylation analysis (DMA) using limma [36] was used to investigate differential methylation patterns between SOS-positive (n = 10) and SOS-negative cheetahs (n = 28). The DMA was limited to liver samples from individuals over 3 years old (n = 8) and included a total of 30 samples.

The DMA results indicated 4,269 differentially methylated sites in SOS-positive versus SOS-negative cheetahs ($p < 10^{-4}$, p.adj < 0.05) (S11 Table), with 3,564 significantly down-methylated (hypomethylated) CpG sites and 705 significantly up-methylated (hypermethylated) CpG sites, measured by log2 fold change (logFC) (Fig 4A). The top four significantly differentially methylated CpGs included cg07680280 in *BCL11A* intronic region (logFC = −0.24), cg15022739 in *CELF4* downstream intergenic region (logFC = −0.15), cg00933498 in *EPHA7* downstream intergenic region (logFC = −0.167), and cg13260846 in *CLIC5* upstream intergenic region (logFC = −0.16) (Fig 4B). The genomic coordinates of probe-aligned genes were used to build a Manhattan plot to portray the distribution of the SOS-related CpG sites across the feline genome (Fig 4C). SOS-related CpGs were located in genic and intergenic regions, and the proportion of hypomethylated versus hypermethylated CpGs was higher across all regions (Fig 4D).

This pilot differential methylation SOS analysis revealed 3,564 hypomethylated and 705 hypermethylated significant CpG sites in SOS-positive individuals compared to SOS-negative individuals. Heatmap visualization and dendrogram clustering of the top 100 CpG sites indicated distinct methylation patterns in most SOS-positive liver samples: eight out of ten SOS-positive samples clustered distinctly away from the SOS-negative samples and displayed methylation patterns that differed between the two groups (Fig 5A).

We also visualized the seven blood samples on the heatmap to determine whether SOS-related methylation patterns could be detected at these CpGs in blood prior to detection on necropsy. Of the seven blood samples tested, two corresponded with liver samples of cheetahs diagnosed with SOS, collected five years before the cheetahs' death in both cases. These blood and liver samples are outlined and connected in black to indicate correlation (Fig 5B). The heatmap revealed two major clusters: one predominantly comprising non-SOS samples (left) and the other enriched for SOS samples (right). Within the non-SOS cluster, the blood samples clustered away from the liver samples.

Kyoto Encyclopedia of Genes and Genomes (KEGG) pathway enrichment analysis on genes associated with the differentially methylated CpGs in SOS cases was performed (Fig 6A-C). Analysis of all CpG-associated genes (Fig 6A), as well as separate analyses on hypermethylated (Fig 6B) and hypomethylated CpGs (Fig 6C), was conducted to discern methylation alterations and their potential implications.

## Discussion

This study presents robust and accurate epigenetic clocks for age and sex prediction in cheetahs using the HorvathMammalianMethylChip40 and biobanked tissues. This Illumina array platform contains probes for 37,492 CpG sites that are conserved across mammalian species and tissues, allowing for pan-species and pan-tissue applications. Raj et al validated a feline-specific clock using blood samples from healthy domestic house cats [11]. We aimed to build on this study using cheetah liver tissue samples collected from individuals with available biomedical data and known ages to build a more precise, cheetah-specific epigenetic clock, as well identify potential biomarkers of cheetah health by focusing on sinusoidal obstruction syndrome (SOS).

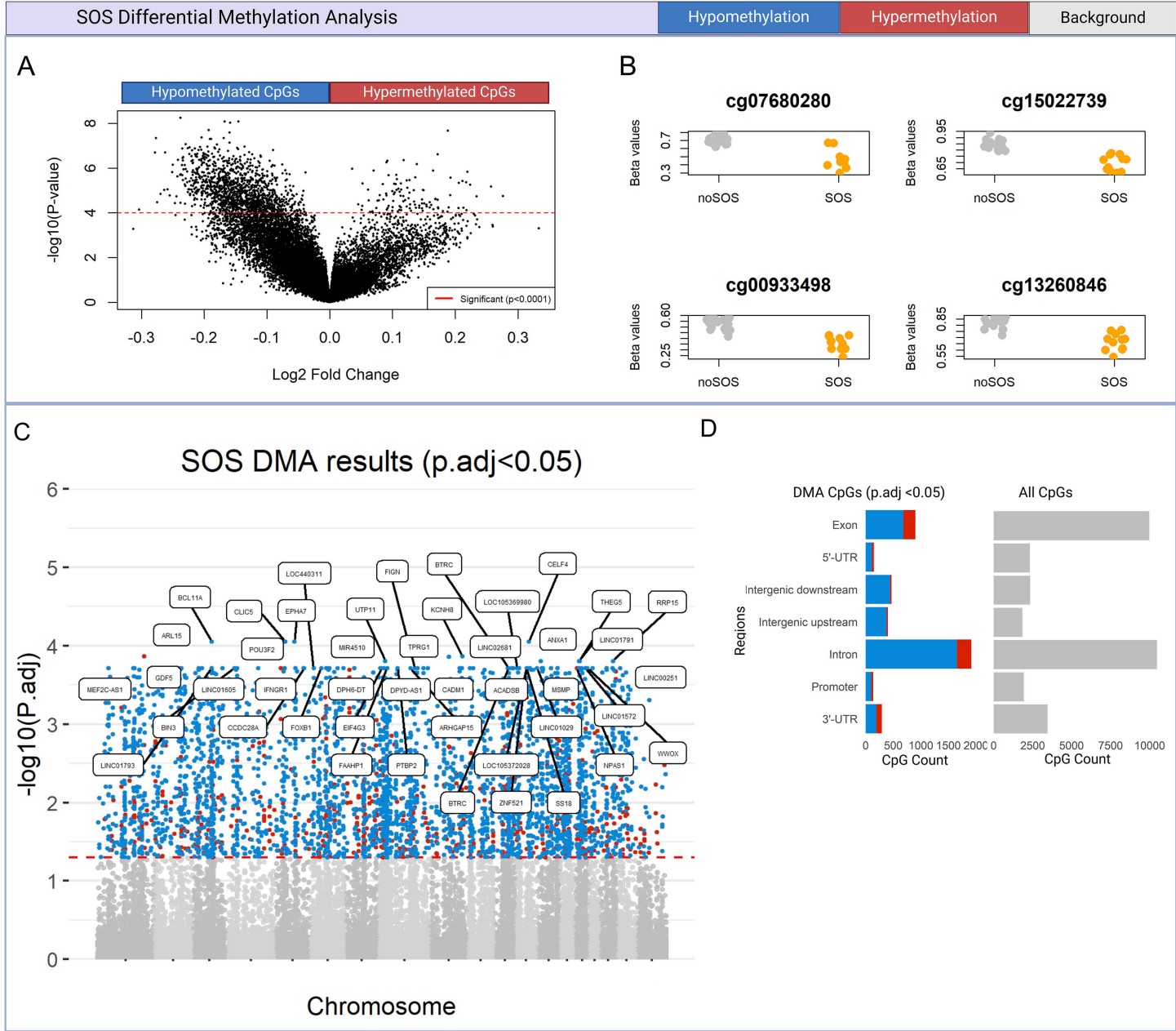

**Fig 4. DMA results.** Results of the DMA conducted using limma [36] on SOS-positive vs SOS-negative samples. **A)** Volcano plot displaying CpG sites that exhibit hypo- or hyper-methylation in SOS-positive individuals compared to SOS-negative individuals (n = 30). Values to the left of 0.0 represent hypomethylated CpGs (log2 fold change < 0), and values to the right of the 0.0 represent hypermethylated CpGs (log2 fold change > 0). Y-axis indicates -log10 transformed p-values from the DMA: higher values portray lower p-values. The red line indicates significance threshold p = 10e-4. **B)** The methylation values (beta values) for the top four differentially methylated CpGs. **C)** Manhattan plot of the DMA liver samples (n = 38) with each point representing a CpG site. CpG genomic coordinates are estimated from the alignment of array probes to the feline genome (Felis_catus_9.1.100). Significant CpGs (p.adj < 0.05) are highlighted to depict direction of association: red indicates hypermethylation (z > 0), and blue depicts hypomethylation (z < 0). CpGs with lowest p.adj values are labeled by adjacent genes. **D)** Location of significant CpGs (p.adj < 0.05) relative to adjacent transcriptional start site, with red indicating hypermethylated CpG proportion and blue indicating hypomethylated CpG proportion. Gray bars in the right graph indicate location of the total 34,851 array probes that map to the feline genome (Felis_catus_9.1.100).

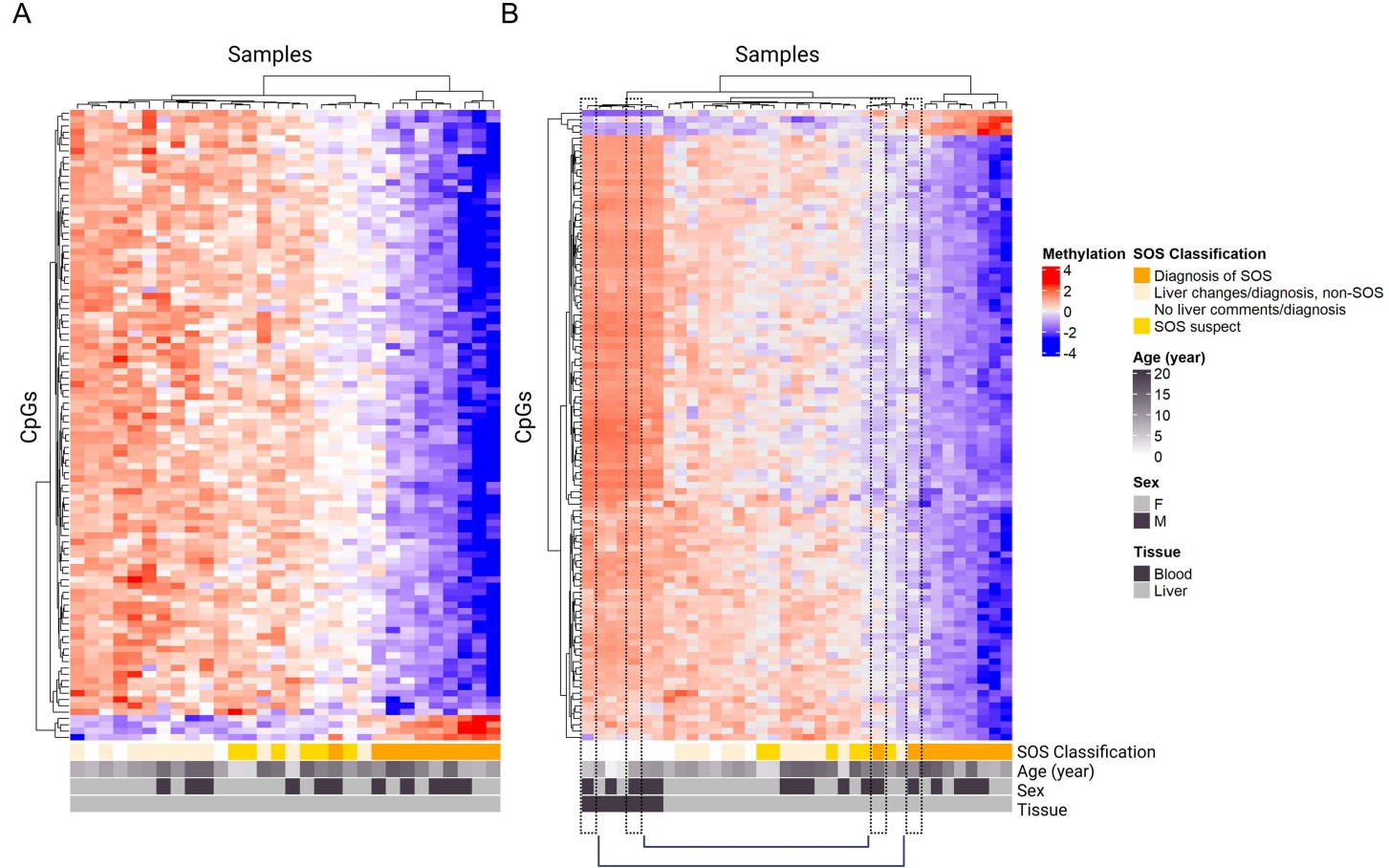

**Fig 5. DMA heatmap analysis.** Heatmap of methylation patterns across liver and test blood samples for selected CpG sites from the DMA comparing the top 100 significant CpG sites. **A)** Heatmap of methylation patterns in liver samples. **B)** Heatmap of methylation patterns in both liver and tested blood samples. The top dendrogram indicates unsupervised hierarchical clustering of samples, and the dendrogram on the left indicates unsupervised hierarchical clustering of CpG sites. Heatmap coloring is determined by Z-score, where colors represent the divergence of a particular gene in a particular sample, as compared to the mean value for that gene over all samples. Red shows increased methylation patterns (hypermethylation), and blue shows decreased methylation patterns (hypomethylation). The key below the heatmap indicates SOS status, age, sex, and tissue. The first row portrays SOS status, with orange indicating diagnosis of SOS at death, yellow indicating SOS-suspect, light-yellow indicating non-SOS liver changes/diagnosis, and white indicating no liver comments/diagnosis. Age spectrum is portrayed in gray, with lighter gray indicating younger age and darker gray indicating older age. Sex is portrayed in gray, with light gray indicating female and dark gray indicating male. Tissue is depicted in the bottom trait bar, with light gray indicating liver and dark gray indicating blood. Correlated samples are indicated with dashed black lines.

## Age and sex biomarkers

The domestic cat clock performed moderately well but had greater than 20% variance with our cheetah data when predicting age in various tissue types and ages. In this study we produced more accurate cheetah-specific clocks that addresses these shortcomings in cheetah age prediction.

Biobanked cheetah blood samples were limited compared to tissue samples (e.g., liver) collected at necropsy. We investigated whether a clock built on both liver (SDZWA) and blood (MCDB) samples would allow for age prediction across various tissue types (SDZWA liver, blood and skin). Pan-tissue clocks for age and sex prediction are particularly useful in wildlife studies where sample availability can vary: while blood samples are sometimes collected from live

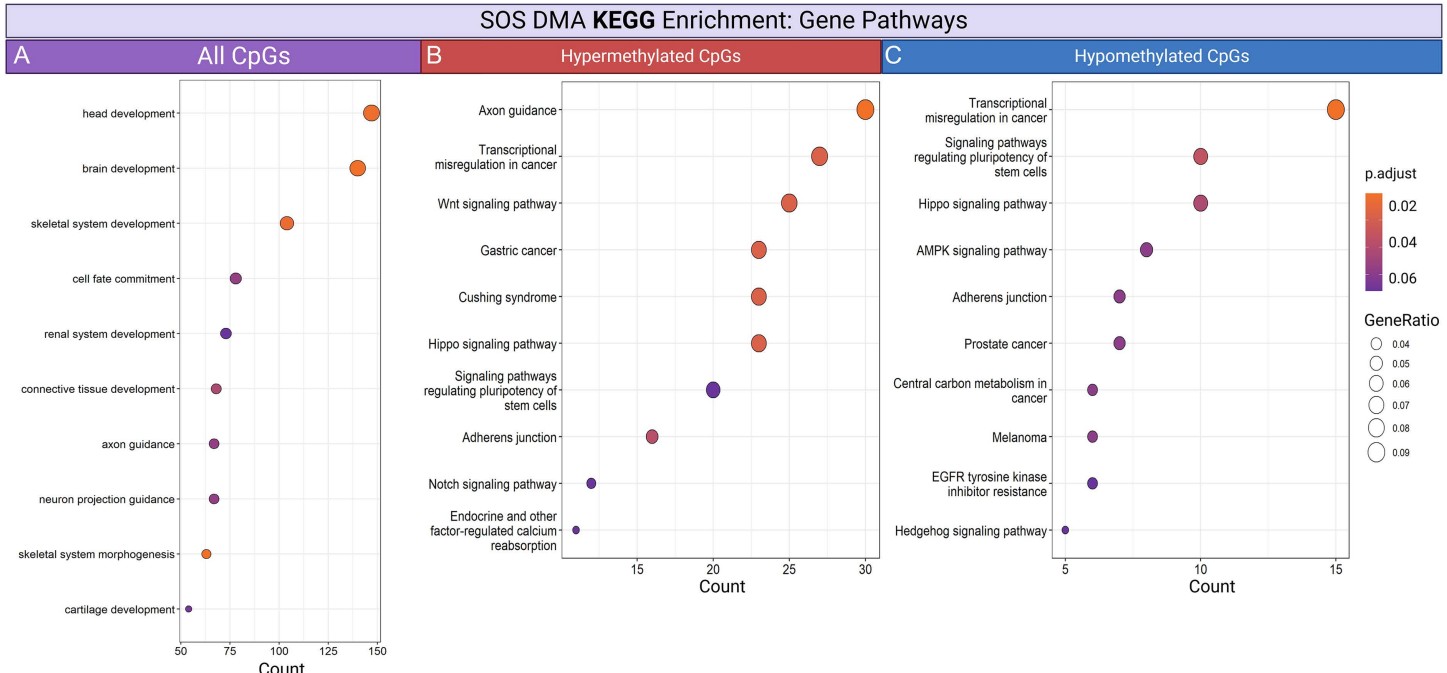

**Fig 6. DMA pathway enrichment analysis.** Gene-level pathway enrichment analysis of significant CpGs from the DMA using human Hg38 annotations of CpG sites. **A)** Includes all pathways with significant CpG sites, **B)** only hypermethylated pathways, and **C)** hypomethylated pathways. The color coding represents adjusted p-value, where darker colors indicate higher significance. Point size reflects the gene ratio. Background genes were limited to the 34,851 array probes that map to the feline genome (Felis_catus_9.1.100). KEGG pathway enrichment analysis of significant CpG sites. The x-axis shows the number of genes involved in each pathway, and the y-axis lists the significant KEGG pathways.

animals, tissues such as liver or skin are also collected from deceased animals in forensic or surveillance contexts. Therefore, species-specific pan-tissue clocks can enhance the utility of age prediction models.

The CheetahClock, trained on methylation profiles from our liver samples and cheetah blood data from MCDB, accurately predicted age within one year for five out of seven cheetah blood test samples aged 5.6–10.2 years. These samples represent the adult age range for wild cheetahs, with true ages estimated within 9 months. Improved age estimates enable conservation teams to make informed management decisions, such as identifying animals with remaining breeding potential, larger ranges, or higher disease risk requiring enhanced monitoring. The model also utilized biobanked tissues stored for up to 38 years (S1 Table), highlighting the value of retrospective epigenetic age studies.

In two samples from our blood test set, CheetahClock overestimated ages (e.g., predicting 5.89 years for an actual age of 2.5 years). The CheetahClock also showed a slight tendency to overestimate ages for MCDB cheetah blood samples in the LOOCV analysis around ASM (e.g., actual age = 1.0 year, predicated age = 3 years). We tested a range of ages of sexual maturity (ASM) (2–3 years) for log-linear transformation and found similar overestimations, suggesting inaccuracy stems from accelerated aging patterns that occur around the ASM that the CheetahClock does not fully capture. Predictions from the CatClock by Raj et al., which does not utilize age transformation, also overestimated age in these two samples [11].

The log-linear transformation used in the CheetahClock to account for increased methylation around ASM may not fully capture the non-linear dynamics of methylation changes occurring during this developmental period, which could lead to age prediction errors in individuals approaching or reaching ASM. Metabolic studies of captive cheetahs have shown they exhibit a rapid, nonlinear phase of physiological development as they approach sexual maturity (roughly 1.5–3

years old) [41,42]. For example, male cheetahs reach adult weight by around 21 months, and their testosterone concentrations increase to reach adult levels by 18–24 months [42]. Female cheetahs likewise attain adult body weight around 21 months, and between 24–30 months their estrogen levels increase and they begin ovarian cycling [41]. In addition to these sex hormone shifts, glucocorticoid levels also become elevated during this pubertal transition [41,42]. These nonlinear developmental trajectories suggest that DNA methylation may also change rapidly and irregularly during this life stage.

Other epigenetic clock studies have found deviations from log linear transformation, such as square root transformation in primates, leads to better predictions in animals before ASM [43]. Duan et al. suggest that methylation dynamics are often exponential, with non-linear changes occurring over time [44]. Additionally, nonlinear dynamics during transitions from early to mid-life and mid-to-late life, have been observed in other species and suggest a more programmed, stage-based aging process [45]. Our findings support these patterns, particularly around ASM, suggesting methylation changes occur more rapidly or irregularly during early adulthood. In cheetahs, accelerated aging or abrupt changes in methylation around ASM may reflect such nonlinear dynamics, which are not fully captured by our current model. It is also possible that sex-specific differences in pubertal timing contribute to the observed overestimations around ASM, particularly if methylation shifts occur earlier or more abruptly in one sex. Expanding our approach to account for these shifts, possibly by incorporating a stage- and sex-based classifier and adding additional samples to our training set around ASM, could better capture the biological complexities of aging and improve prediction accuracy for younger individuals.

Additionally, our test set included biobanked whole blood samples rather than isolated cell populations. Variation in blood cell composition, particularly in individuals with subclinical disease or immune activation, may have introduced methylation variability and affected prediction accuracy [46]. Future studies may benefit from incorporating blood cell composition analysis or using sorted cells to improve disease and age models.

Despite the high correlation ($r = 1$) observed between age predictions in skin samples produced by CheetahClock, the median absolute error (MAE) was 1.77 years, though this is a slight improvement from UniversalClock2 predictions ($r = 0.95$, MAE = 1.99). EWAS analysis of age in skin showed distinct methylation patterns, including more hypomethylation in promoter and 5'UTR regions with age compared to liver and blood. Promoter and 5'UTR regions are key transcription regulatory regions where DNA methylation is often associated with transcription suppression, and increased methylation of CpGs in these regions with age has been observed in blood [11,47,48]. Skin may exhibit different methylation dynamics due to its unique environmental exposure and regenerative properties, which could lead to different epigenetic aging processes compared to internal organs like the liver and blood [49]. Although skin biopsies were typically collected from the trunk or abdomen in these necropsy cases, we cannot confirm that all samples originated from the same anatomical region. Variation in sampling location could introduce additional variability, as skin from different body parts may experience different environmental exposures or aging rates. Incorporating more anatomically consistent skin samples into future studies would capture more tissue-specific methylation patterns, potentially improving the MAE while maintaining high correlation.

CheetahClock predictions from LOOCV and test samples demonstrate its robustness for cheetah age prediction across tissues, including clinically available samples like blood and skin. Because the model was developed using samples from zoo-housed individuals, its accuracy in wild populations may be influenced by differences in diet, stress and veterinary care which could affect epigenetic aging. Additional testing in wild individuals with known or estimated ages will be important to evaluate and calibrate for these potential differences. While further validation with field-curated samples is needed, this is a great advancement for aging these felines in the wild that may potentially replace current expensive and invasive methods of cheetah age approximation such as radiograph analysis of skull morphology and tooth pulp cavity analysis [17].

CheetahClock predictions for other exotic feline species, including tigers and lions, showed high correlation and high but consistent MAE, suggesting that while age-related CpG patterns are similar, the clock is best calibrated for cheetahs. We developed FelidClock by training on a combined dataset of cheetah, lion (n = 7), and tiger (n = 8) profiles, using 46

CpGs (33 overlapping with CheetahClock). LOOCV analysis showed effective calibration for multiple species, with similar accuracy in cheetah blood and skin samples, suggesting FelidClock could age three big-cat species in field studies within 1–2 years of actual age on blood, skin and liver samples.

We also built a sex prediction model using the combined liver and blood training dataset that accurately predicted sex in all LOOCV and cheetah, tiger and lion test samples. The pan-mammalian clock sex predictions were also accurate across our samples, indicating DNA methylation is a highly reliable tool for sex prediction.

Epigenome-wide association studies (EWAS) were performed across liver, blood, and skin samples, with the largest number of significant CpGs identified in liver and blood due to their higher sample sizes. CpGs linked to genes previously associated with aging in mammals, such as *NEUROD1*, *SALL1*, and *NR2F2*, were significantly associated with age in these tissues [11,33]. Although the exploratory EWAS of the smaller skin sample set yielded limited findings, it identified 79 overlapping CpGs between liver and skin, four between blood and skin, and four across all three tissues (*NEUROD1*, *TNR*, *C1D*, and *UNCX*), indicating shared age-related epigenetic markers. Including liver samples in the training set for CheetahClock appeared to enhance its predictive power in skin samples compared to clocks that relied solely on blood (e.g., CatClock). The 79 overlapping CpGs between liver and skin, compared to four overlapping between blood and skin, suggest that the liver samples contribute significant age-related epigenetic markers that may improve the model's accuracy across different tissues.

These methylation-based sex and age prediction tools will aid in conservation efforts by providing accurate demographic profiling for wildlife population studies and epidemiological investigations; for example, when samples are collected from the environment (hair snares, fecal) or the animal in hand does not have sexual dimorphisms. Future research aimed at applying these models to diverse sample types such as hair or fecal samples can increase the potential for less-invasive sampling and broader population coverage. Furthermore, the identification of a relatively small number of species-specific CpG sites associated with age and sex facilitates the development of field-applicable protocols such as bisulfite-PCR or qPCR that could reduce costs, enabling scalable applications for wildlife management [15,50]. This especially true when paired with portable, cost-effective sequencing tools that can be deployed in global decentralized laboratories for broad applications [51].

## SOS biomarkers

In addition to age biomarkers, the epigenome may also contain health biomarkers that could be used to track health parameters and disease status. DNA methylation patterns may manifest prior to observable clinical alterations, and detecting disease-related methylation patterns could facilitate early disease detection and intervention [52]. As epigenetics functions dynamically at the interplay between genetics and the environment, DNA methylation also offers potential mechanistic links between environmental perturbations or exposures to environmental contaminants, and disease manifestation. For example, in plants [53], birds [54], livestock [4,55], and rodents [56], DNA methylation has been shown to regulate stress pathways in response to environmental stimuli such as excessive heat and toxicant exposure. Investigating DNA methylation patterns that correlate with disease status could identify critical pathways involved in disease progression and potentially provide holistic biomarkers for tracking animal health. By building tools that can evaluate chronological age and health status, we can begin to monitor health in species we previously had a limited ability to monitor.

Cheetahs present a unique opportunity to study DNA methylation biomarkers due to their remarkably homogeneous genomes from a population bottleneck approximately 10,000 years ago [57]. This genetic uniformity minimizes confounding genetic factors, allowing us to attribute epigenetic variations to environmental or stochastic influences and identify conserved epigenetic patterns [58]. One clinically significant condition in captive cheetahs is sinusoidal obstruction syndrome (SOS), characterized by idiopathic fibrosis targeting vessels that can lead to vascular occlusion, necrosis, and potential liver failure [22,59–61]. While studies have focused on captive populations, SOS has also been observed in wild cheetahs, as well as in other species like snow leopards and humans, though the exact causes in cheetahs remain unclear

[59,62–64]. Between 2015 and 2017, an outbreak of SOS occurred at the San Diego Zoo Safari Park without identifiable causes [65].

We conducted a pilot analysis using Horvath Array methylation profiles on biobanked tissues from SOS-positive and SOS-negative cheetahs. All but two SOS-positive liver samples clustered distinctly from non-SOS cases. Notably, these two cheetahs were not diagnosed with SOS while they were alive, and SOS was an incidental finding in both cases during necropsy and reported as "mild". The methylation patterns of these two samples were similar to SOS-suspect samples. Other liver pathologies did not show this clustering pattern, suggesting that SOS-suspect samples might have been undergoing similar biological processes or changes to SOS-positive cases. These findings highlight the potential of methylation biomarkers in liver biopsies for identifying SOS pathology before clinical symptoms appear.

We analyzed the methylation of six blood samples to determine whether SOS-related methylation patterns could be detected in blood several years prior to SOS diagnosis at death. At the time of blood collection, the SOS disease status of these cheetahs was assumed negative. SOS disease status was not visually distinct on the heat map based on the two blood samples from known "mild" SOS-positive cases at death. As previously stated, these two liver samples showed clustering away from the other SOS-positive samples, clustering closer to the SOS suspect and other liver disease/non-SOS samples. Thus, from this analysis it is not clear if severe SOS-positive cases and related methylation patterns can be observed in blood samples compared to liver samples either prior to or at the time of liver biopsy diagnosis, or if a tissue-specific SOS-blood model is needed.

The DMA and heatmap analysis revealed that most CpGs implicated in SOS-positive samples were hypomethylated compared to SOS-negative samples, while some sites exhibited distinct hypermethylation. DNA methylation has been studied more recently in its relation to hepatic fibrosis, specifically its influence of hepatic stellate cells (HSC) which are central to liver fibrosis [66–68]. Hypomethylation at genes implicated in the DMA like *TGF-β* can drive HSC activation, promoting fibrosis-related protein expression (e.g., collagen) and extracellular matrix accumulation, exacerbating fibrosis [67,69,70]. Hypermethylation of *SMAD7*, also observed in our analysis, is linked to HSC activation, highlighting the roles of both hypo- and hypermethylation in fibrosis regulation [67]. In the context of SOS, similar aberrant methylation patterns could contribute to fibrotic responses within the liver and lead to SOS progression.

KEGG enrichment analysis of genes associated with CpGs from the DMA indicated pathways such as Hippo signaling, AMPK signaling, Wnt signaling, Notch signaling, and endocrine- and factor-regulated calcium reabsorption are potentially implicated in SOS in cheetahs. These pathways are known to regulate cellular processes like HSC activation. The Wnt pathway, for instance, is implicated in cell proliferation and fibrosis through their influence on HSCs [71–77]. Hippo pathway elements *YAP* and *TAZ* are key regulators of HSC activation, and AMPK activation has been shown to inhibit *YAP/TAZ* signaling [78–81]. Notch signaling is also a critical regulator in the development of liver fibrosis, primarily by promoting the activation of HSCs in response to liver injury, and also affects other liver cell types, like liver sinusoidal endothelial cells and hepatocytes, where it influences the expression of HSC-activating genes [82,83]. Similarly, calcium signaling plays a vital role in liver regeneration by facilitating calcium-dependent interactions for HSC function [66].

The present findings suggest there are SOS pathogenesis-related methylation patterns present in our samples, and initial exploratory analyses on differentially methylated CpG sites and their associated genes produced results that warrant further exploration. The array used in this study mainly contained human-aligned, conserved CpG sites related to age, and a more inclusive analysis such as a genome-wide methylation study may uncover many more CpG regions and genes related to cheetah SOS. Further investigation of these interconnected pathways, including multi-omic analyses integrating gene expression and methylation data, is essential to understanding their contribution to SOS and identifying diagnostic biomarkers and therapeutic targets. While no definitive SOS etiology conclusions can be drawn from methylation data alone, this study presents a foundation for further exploration into the relationship between DNA methylation, gene expression, and fibrotic pathways associated with the progression of SOS.

## Conclusion

In this study, we validate the use of novel epigenetic clocks for age and sex prediction in cheetahs, utilizing biobanked samples from clinical and necropsy collections. These tools enhance individual and population-level studies of wild and captive cheetahs by providing a more feasible alternative to current aging methodologies that are often difficult or impractical in field settings. Our findings reveal age and sex-related biomarkers in methylation profiles derived from both freshly collected and biobanked liver, blood, and skin samples, laying the groundwork for future research aimed at refining these methods for practical application in both field and clinical environments. Additionally, we present a pilot analysis of epigenetic markers of health, specifically concerning cheetah sinusoidal obstructive syndrome (SOS). By identifying species-specific biomarkers of age, sex and health, we can leverage techniques such as bisulfite PCR and portable sequencing technology to monitor and manage zoo-housed and wild animal populations.

## Supporting information

**S1 Fig. Unsupervised hierarchical clustering on liver, blood, and skin samples.** Unsupervised hierarchical clustering was performed to detect outliers in methylation profiles for **A)** liver, **B)** skin, and **C)** blood samples. The red lines in each dendrogram represent the cutoff heights, chosen based on the point of distinct separation in the clustering structure. Samples that clustered separately above this cutoff were identified as outliers and excluded from further analysis. **D)** Unsupervised hierarchical clustering of all cheetah samples used (outliers removed with the trait bars for annotation. Age spectrum is portrayed in red, with lighter red indicating younger age and darker red indicating older age. Sex is portrayed in pink and blue, with pink indicating female and blue indicating male. Tissue is depicted in red, orange and tan, with red indicating blood, orange indicating skin, and tan indicating liver. Study is portrayed in green and dark brown, with green indicating samples from SDZWA and dark brown indicating samples from MCDB. The DNA extraction kit used is indicated in blue and red, with blue indicating Qiagen extraction kit, red indicating NEB Monarch extraction kit, and grey indicating unknown (MCDB samples).
(TIFF)

**S2 Fig. Universal Pan-mammalian clock results.** Evaluation of Universal Pan-mammalian clocks on SDZWA cheetah methylation profiles. **A)** PanClock3 predicted age versus chronological age for cheetah blood, liver and skin samples. **B)** PanClock2 predicted age versus chronological age for cheetah blood, liver and skin samples. **C)** UniversalClock2Skin predicted age versus chronological age for cheetah skin samples. **D)** CatClock predicted age versus chronological age for cheetah blood samples. **E)** CatClock predicted age versus chronological age for cheetah blood, liver and skin samples.
(TIF)

**S3 Fig. Effect of different ASM on CheetahClock.** Evaluation of different ages of sexual maturity (ASM) on Cheetah-Clock prediction results. **A-C)** Evaluation of ASM = 1.5 on **A)** training liver and blood samples, **B)** test blood samples, **C)** test skin samples. **D-F)** Evaluation of ASM = 3 on **A)** training liver and blood samples, **B)** test blood samples, **C)** test skin samples.
(TIF)

**S1 Table. Sample metadata.** Table with details on all samples used in the study, including both SDZWA and MCDB samples: individual name, sample name, tissue type, age at collection, sex, DNA extraction date, DNA extraction kit used, and storage time (date extracted – date collected).
(CSV)

**S2 Table. CheetahClock age coefs.** List of CpGs and coefficients used in CheetahClock for age prediction.
(CSV)

**S3 Table. CheetahClock age LOOCV.** Results from CheetahClock for age leave-one-out cross-validation (LOOCV) on liver and blood samples.
(CSV)

**S4 Table. CheetahClock age test.** Results from CheetahClock for age applied to test set of blood and skin samples.
(CSV)

**S5 Table. CheetahClock sex coefs.** List of CpGs and coefficients used in CheetahClock for sex prediction.
(CSV)

**S6 Table. CheetahClock sex LOOCV.** Results from CheetahClock for sex leave-one-out cross-validation (LOOCV) on liver and blood samples.
(CSV)

**S7 Table. CheetahClock sex test.** Results from CheetahClock for sex applied to test set of blood and skin samples.
(CSV)

**S8 Table. FelidClock coefs.** List of CpGs and coefficients used in FelidClock for age prediction across feline species (cheetah, tiger, lion).
(CSV)

**S9 Table. FelidClock age LOOCV.** Results from FelidClock for age leave-one-out cross-validation (LOOCV) on liver and blood samples (cheetah, tiger, lion).
(CSV)

**S10 Table. FelidClock age test.** Results from FelidClock for age applied to test set of cheetah blood and skin samples.
(CSV)

**S11 Table. SOS DMA annotated results.** Results from the differential methylation analysis (DMA) conducted on SOS-positive samples. Includes DMA results for significant CpGs (p.adj < 0.05) found in the analysis: log-fold change (logFC), p-value (P.Value), and adjusted p-value (adj.P.Val), as well as human, cheetah and domestic cat gene annotations for each CpG site and CpG location.
(CSV)

**S1 Data. DNA methylation dataIncludes the sample metadata sheet, raw Illumina IDAT files, and normalized beta value CSV file generated from the HorvathMammalMethylChip40 array.** Data are publicly available on Gene Expression Omnibus (GEO) under accession number GSE310779.
(ZIP)

**S1 Document. Inclusivity in Global Research Questionnaire.** PLOS Inclusivity in Global Research Questionnaire outlining ethical approvals, sample collection details, and authorship considerations in this study.
(DOCX)

## Acknowledgments

We thank the pathology, histology, and Conservation Genetics research teams at the San Diego Zoo Wildlife Alliance, Dr. Patricia Gaffney, Dr. Ilse Stalis and Dr. Carmel Whitte for their guidance. We are also grateful to the molecular diagnostics team at the San Diego Zoo Wildlife Alliance for their support, and to Andy Richardson, Dr. Candace Williams and Dr. Mrinalini Erkenswick Watsa for their assistance with bioinformatics.

## Author contributions

**Conceptualization:** Caroline E. Moore.

**Data curation:** Michelle Cristi Ysrael.

**Formal analysis:** Michelle Cristi Ysrael.

**Funding acquisition:** Caroline E. Moore.

**Investigation:** Michelle Cristi Ysrael.

**Project administration:** Steven Kubiski, Caroline E. Moore.

**Resources:** Steven Kubiski, Caroline E. Moore.

**Supervision:** Steven Kubiski, Caroline E. Moore.

**Validation:** Michelle Cristi Ysrael.

**Visualization:** Michelle Cristi Ysrael.

**Writing – original draft:** Michelle Cristi Ysrael.

**Writing – review & editing:** Steven Kubiski, Caroline E. Moore.

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
