## [Decision Letter · Decision Letter 0]

16 Jun 2025

Dear Dr. Ysrael,

Thank you for submitting your manuscript to PLOS ONE. After careful consideration, we feel that it has merit but does not fully meet PLOS ONE’s publication criteria as it currently stands. Therefore, we invite you to submit a revised version of the manuscript that addresses the points raised during the review process.

Appropriate references should be cited to support the description of the liver disease—veno-occlusive disease (VOD), also known as sinusoidal obstruction syndrome (SOS)—in cheetahs. Notably, SOS is relatively common in captive cheetahs but occurs much less frequently in free-ranging individuals. This distinction should be clearly acknowledged, as it limits the applicability of the epigenetic clock developed in this study to wild cheetah populations. The following references are relevant and should be cited to support this point:

 https://doi.org/10.1177/030098588802500107

 https://doi.org/10.1638/2020-0084

The authors should provide a clear justification for the use of a human-based methylation array for cheetahs, particularly given the availability of species-specific genomic resources. To enhance the biological relevance and interpretability of the findings, the authors should evaluate the homology of associated genes between cheetahs and humans and confirm the genomic positions of the identified CpG sites by mapping them to the cheetah reference genome. This additional analysis would provide stronger support for the study’s conclusions.

publication criteria  and not, for example, on novelty or perceived impact.

We look forward to receiving your revised manuscript.

Kind regards,

Purnima Singh, PhD

Academic Editor

PLOS ONE

3. In the online submission form, you indicated that your data will be submitted to a repository upon acceptance.  We strongly recommend all authors deposit their data before acceptance, as the process can be lengthy and hold up publication timelines. Please note that, though access restrictions are acceptable now, your entire minimal  dataset will need to be made freely accessible if your manuscript is accepted for publication. This policy applies to all data except where public deposition would breach compliance with the protocol approved by your research ethics board. If you are unable to adhere to our open data policy, please kindly revise your statement to explain your reasoning and we will seek the editor's input on an exemption.

Additional Editor Comments:

The authors should provide access to the raw data through a publicly available repository. This is essential to allow independent validation of the results and to ensure compliance with journal requirements.

Reviewers' comments:

Reviewer's Responses to Questions

**Comments to the Author**

1. Is the manuscript technically sound, and do the data support the conclusions?

Reviewer #1: Yes

Reviewer #2: Yes

2. Has the statistical analysis been performed appropriately and rigorously?

Reviewer #1: Yes

Reviewer #2: Yes

3. Have the authors made all data underlying the findings in their manuscript fully available?

Reviewer #1: No

Reviewer #2: No

4. Is the manuscript presented in an intelligible fashion and written in standard English?

Reviewer #1: Yes

Reviewer #2: Yes

Reviewer #1: This study on methylation changes in cheetahs is novel and valuable, primarily due to its utilization of a unique and novel dataset: biobanked samples from the San Diego Zoo. The availability of precise age and health records for these zoo housed animals is a significant strength. The authors successfully developed a highly accurate epigenetic clock specifically for cheetahs.

Interestingly, the study also presents a novel felid epigenetic clock applicable to both tigers and lions. Further, the authors introduce a new methylation based predictor of sex. A particularly exciting aspect of this research is the exploration of methylation analysis utility in informing disease investigations in zoo housed cheetahs, with a specific focus on sinusoidal obstructive syndrome. Although the latter analysis is only a pilot study, it will be of interest to many readers. The statistical analysis appears robust. The article is well-written but I suggest to add some citations, as detailed below.

I have only minor comments.

1) Optional Suggestion: I wonder if the accuracy of the felid clock could be enhanced by incorporating blood methylation samples from domestic cats. Such data are available through the Mammalian Methylation Consortium. I recommend citing the following article, which presents a clock for domestic cats: K. Raj et al. (2021). GeroScience. Available at: https://pmc.ncbi.nlm.nih.gov/articles/PMC8599556/

2) Line 135: Please add a citation to Horvath (2013, Genome Biology) after the sentence "Log-linear age transformation was performed to account for 'accelerated aging' methylation patterns observed before age of sexual maturity (ASM)." This citation is appropriate as Horvath's work introduced the log-linear transformation for human epigenetic clocks.

3) Data Availability Statement: It would be desirable to deposit the data on the Gene Expression Omnibus (GEO) webpage, including the relevant genomic platform number=GPL28271, to ensure wider accessibility.

Reviewer #2: Reviewer´s Comments for manuscript submitted to PLOS ONE

The manuscript entitled “Investigating epigenetic biomarkers of age and health in South African cheetahs (Acinonyx jubatus jubatus)” presents an important and timely study on the development of an epigenetic clock for cheetahs, a valuable tool for wildlife conservation and management. The authors have generated compelling results showing clear age prediction across different tissues and have extended the analysis to explore biomarkers for sex and Sterile Osteosclerotic Sclerosis (SOS). The study is a significant contribution to the field of conservation genomics. While the core findings are strong, the manuscript would be substantially improved by addressing several key areas. The primary concerns relate to the overall narrative structure, the justification for the experimental design, the discussion of the study's limitations (particularly the use of captive animals and a human-based microarray), and the need for greater methodological detail. The following suggestions are intended to help the authors strengthen the manuscript for publication. I recommend Major Revisions.

Strengthen the Central Narrative and Hypotheses:

The manuscript currently presents analyses of age, sex, and SOS disease, but the biological rationale connecting these variables is not fully developed. The manuscript would benefit from a clearer articulation of the overarching research questions or hypotheses at the end of the Introduction. This would help frame why these specific variables were chosen and what the expected outcomes were, thereby improving the manuscript's logical flow and structure.

Title Revision: The current title does not fully reflect the scope of the investigation. "Health" is a broad term, whereas the study specifically investigates the disease SOS. Furthermore, the use of exclusively captive individuals is a critical detail for interpreting the results, especially those related to biological age. I suggest revising the title to be more specific, for example: “Investigating epigenetic biomarkers of age, sex, and disease in captive South African cheetahs (Acinonyx jubatus jubatus)”.

Applicability and Limitations of Using Captive Animals: The study's reliance on captive individuals is a significant limitation when considering the application of this clock to wild populations. The Discussion should more explicitly address this. Please discuss how factors unique to captivity (e.g., diet, stress levels, veterinary care) might influence epigenetic aging compared to wild counterparts. A more robust discussion is also needed on the challenges of accounting for accelerated aging in wild individuals whose life histories are unknown.

Justification for Genomic Methods: The study utilizes the HorvathMammalianMethylChip40, which was designed for the human genome. While useful for cross-species comparisons, the cheetah genome has been well-sequenced. The authors should justify the use of the human chip and, more importantly, validate their findings. I strongly recommend mapping the identified CpG sites to the cheetah reference genome to confirm their locations and assess the homology of any associated genes between humans and cheetahs. This would significantly strengthen the biological interpretation of the results.

Data Availability: A link to the raw data is missing. For transparency and reproducibility, which is a key policy for PLOS ONE, please provide a link to a public repository where the data are available.

Specific Comments

Abstract:

Please remove non-standard abbreviations from the abstract to adhere to journal guidelines.

L24: I suggest changing the terminology from “over the age of maturation” to a more standard term like “adult” individuals, and specifying the age threshold used (e.g., age > 2 years).

L30-33: The connection between these lines and the preceding sentences is unclear. Please revise to improve the flow of logic.

Introduction:

L44: Please specify whether the epigenetic modifications mentioned correlate positively or negatively with age in this context.

L44-46: For clarity and conciseness, I suggest removing the phrase “and commonly occurs in regions of DNA where these cytosine residues are..”.

L47: Please edit "methyl groups" to the standard form "methyl-groups".

L48: Please clarify what is meant by “structural changes” in this context.

L49: A reference is missing.

L51: Please define what is meant by "Pan-species" upon first use.

L61-62: I suggest rephrasing “tools” to something more precise, such as “these traditional age-estimation methods are difficult to apply accurately”.

L63: Please add "Measurements of" or "The study of" before "epigenetic aging" for logical clarity.

L68: It is crucial information that all samples are from captive individuals with known chronological age, sex, and health status. Please state this clearly and early in the Introduction and Methods, as it is currently difficult for the reader to ascertain.

The Introduction is missing an explicit mention of sex as a variable being investigated. Please incorporate this.

L75-81: This section is the ideal place to formulate the clear hypotheses or research questions mentioned in the Major Revisions.

Methods:

L83: It would be very helpful to state explicitly at the beginning of this section that "All animals were born and raised in captivity and held under human care until death," or similar phrasing.

L89: Please change “die” to “died”.

L91: Please clarify if n=7 refers to the number of samples or individuals.

L93-94: I suggest rephrasing; "depicts correlations" could be changed to "indicates repetitive sampling from the same individuals over time" if that is the intended meaning.

L103: Since two different extraction kits were used, please confirm whether a potential batch effect was investigated and controlled for.

L123: The description of the methods here is too brief; "converted using bisulfite sequencing and put through their previously published pipeline" is not well phrased and insufficient. Please provide a more thorough explanation of the steps involved.

L142: How was the sequencing performed, and how many samples were included in this step?

L148: Please state the significance threshold (e.g., p-value) used for the analysis.

L153: Please consider adding a supplementary table listing the sample IDs, their known health status (especially for SOS), and other relevant metadata for the samples used in the final analyses.

L155-156: The sentence “SOS has been diagnosed in populations as early as 5 years old” is unclear. Does this refer to cheetah populations in general? Please rephrase for clarity. Also, please clarify how the age of these animals was determined.

L156-157: The current phrasing implies causality ("To capture potential epigenetic changes that lead to SOS..."). This is a strong claim. I suggest rephrasing to such as: “To identify potential epigenetic changes associated with SOS…”

L158-165: This section describes various SOS states. For clarity, please focus only on the states that were actually included and analyzed in your study.

L183: Please specify which software or online tool was used to perform the KEGG pathway analysis.

Results:

L189-196: This section appears to describe methods rather than results. Please consider moving it to the Methods section to maintain a clear distinction.

L196-199: Please clarify why stillborn individuals are mentioned if they were not included in the analysis, or remove the mention if it is not relevant.

L202: Please provide the justification for excluding outliers from the analysis.

L204: Please define "pan-tissue" in this context.

L216: Do the "blood" samples refer to whole blood or a specific fraction (e.g., PBMCs)? If whole blood was used, please discuss the potential confounding effect of variable blood cell composition, particularly in the context of disease or infection.

L231: Please use the abbreviation LOOCV consistently after it is first defined.

L246-248: In the text, please specify the sample sizes for each group in this analysis (e.g., N=x males, N=y females; N=x liver, N=y blood).

L249: Please revise the section title to be more descriptive of the results found.

Discussion:

L310: Please ensure the terms "hypomethylation" and "hypermethylation" are used consistently throughout the manuscript and that it is always clear which group is being referenced (e.g., "hypomethylated in SOS-positive individuals compared to controls").

L401: This is a good point. Please expand the discussion of accelerated aging in the context of the cheetah's life history traits.

L409: Suggest changing “under” to “before”.

L411: The lines state, that “Methylation dynamics are often exponential.” Please elaborate on how the results presented in this study support this statement.

L412-417: The observed differences around the age of sexual maturation (ASM) could potentially be confounded by the sex of the animals. Please consider and discuss this possibility.

L425: Were skin samples taken from the same anatomical region on all animals? Different body parts may be exposed to different levels of environmental factors and could exhibit different aging rates. Please clarify or discuss this as a potential limitation.

L507: The sentence ending is incomplete ("hypomethylated in xx"). Please complete it.

**Do you want your identity to be public for this peer review?** For information about this choice, including consent withdrawal, please see our Privacy Policy

Reviewer #1: No

Reviewer #2: No

---

## [Author Response · Author response to Decision Letter 1]

31 Jul 2025

(copied from uploaded Response to Reviewers doc)

Reviewer #1

1) Optional Suggestion: I wonder if the accuracy of the felid clock could be enhanced by incorporating blood methylation samples from domestic cats. Such data are available through the Mammalian Methylation Consortium. I recommend citing the following article, which presents a clock for domestic cats: K. Raj et al. (2021). GeroScience. Available at: https://pmc.ncbi.nlm.nih.gov/articles/PMC8599556/

We did initially consider using the domestic cat samples in our clock study, but ultimately decided that since most of the domestic cat samples were neutered, they were not representative of intact wild and zoo-house felids such as cheetahs. This was supported by clustering analysis of the cheetah and domestic cat samples: we observed distinct clustering based on sex in the domestic cat samples (Species trait bar = purple), while the cheetah samples (Species trait bar = blue) did not display such distinct sex-based clustering (see image in uploaded Response to Reviewers document).

2) Line 135: Please add a citation to Horvath (2013, Genome Biology) after the sentence "Log-linear age transformation was performed to account for 'accelerated aging' methylation patterns observed before age of sexual maturity (ASM)." This citation is appropriate as Horvath's work introduced the log-linear transformation for human epigenetic clocks.

A citation to this paper was added (reference #34): “ Log-linear age transformation was performed to account for “accelerated aging” methylation patterns observed before age of sexual maturity (ASM)[34], as done by Lu et al[35] in building the pan-mammalian epigenetic clocks, and by Peters et al[33] in building bottlenose dolphin epigenetic clocks.” (Line 157)

3) Data Availability Statement: It would be desirable to deposit the data on the Gene Expression Omnibus (GEO) webpage, including the relevant genomic platform number=GPL28271, to ensure wider accessibility.

Our team is currently working to obtain the necessary permissions to upload the raw IDAT files, SESAME-normalized data received from The Clock Foundation, and the associated sample metadata sheet to NCBI GEO. We will provide the accession number once it becomes available.

Reviewer #2

Strengthen the Central Narrative and Hypotheses:

The manuscript currently presents analyses of age, sex, and SOS disease, but the biological rationale connecting these variables is not fully developed. The manuscript would benefit from a clearer articulation of the overarching research questions or hypotheses at the end of the Introduction. This would help frame why these specific variables were chosen and what the expected outcomes were, thereby improving the manuscript's logical flow and structure.

A clear hypothesis that addresses the two main study goals was added to the last sentence of the introduction: “We hypothesized that DNA methylation-based models could accurately predict age and sex in cheetahs, even with a limited sample size, and that incorporating multiple tissue types and related felid species could enhance predictive power and generalizability. Additionally, DNA methylation has been studied for its utility as a biomarker in various human and animal diseases [25,26], and we hypothesized that similar epigenetic differences would distinguish SOS-affected and unaffected cheetahs.” (Lines 84-89)

Title Revision: The current title does not fully reflect the scope of the investigation. "Health" is a broad term, whereas the study specifically investigates the disease SOS. Furthermore, the use of exclusively captive individuals is a critical detail for interpreting the results, especially those related to biological age. I suggest revising the title to be more specific, for example: “Investigating epigenetic biomarkers of age, sex, and disease in captive South African cheetahs (Acinonyx jubatus jubatus)”.

To better reflect the study scope, edited the title to: “Investigating epigenetic biomarkers of age, sex, and disease in captive South African cheetahs (Acinonyx jubatus jubatus)” (Lines 4-5)

Applicability and Limitations of Using Captive Animals: The study's reliance on captive individuals is a significant limitation when considering the application of this clock to wild populations. The Discussion should more explicitly address this. Please discuss how factors unique to captivity (e.g., diet, stress levels, veterinary care) might influence epigenetic aging compared to wild counterparts. A more robust discussion is also needed on the challenges of accounting for accelerated aging in wild individuals whose life histories are unknown.

Added: “CheetahClock predictions from LOOCV and test samples demonstrate its robustness for cheetah age prediction across tissues, including clinically available samples like blood and skin. Because the model was developed using samples from zoo-housed individuals, its accuracy in wild populations may be influenced by differences in diet, stress and veterinary care which could affect epigenetic aging. Additional testing in wild individuals with known or estimated ages will be important to evaluate and calibrate for these potential differences. While further validation with field-curated samples is needed, this is a great advancement for aging these felines in the wild that may potentially replace current expensive and invasive methods of cheetah age approximation such as radiograph analysis of skull morphology and tooth pulp cavity analysis [17]. ” (Lines 464-468)

Justification for Genomic Methods: The study utilizes the HorvathMammalianMethylChip40, which was designed for the human genome. While useful for cross-species comparisons, the cheetah genome has been well-sequenced. The authors should justify the use of the human chip and, more importantly, validate their findings. I strongly recommend mapping the identified CpG sites to the cheetah reference genome to confirm their locations and assess the homology of any associated genes between humans and cheetahs. This would significantly strengthen the biological interpretation of the results.

The mammalian methylation array (HorvathMammalianMethyl40 Chip) was developed from human and mouse methylation arrays, but was redesigned by Arneson et al (Arneson et al. 2022) using a 100-way alignment of 99 vertebrate genomes with the human genome to identify highly conserved CpG sites. To highlight this, we added to the introduction: “Arrays used to build these clocks, including the HorvathMammalMethylChip40 developed by Arneson et al[14], were specifically designed for cross-mammalian application using conserved CpG sites selected through multi-species alignments ” (Lines 52-55)

We edited Supplemental Table 11 (annotations of significant CpGs from the SOS DMA) to include both cheetah (Aci_jub_2) and domestic cat (FC.9.0.100) annotations in addition to the original human (Hg38) annotations. These annotations were part of the cheetah annotation sheet that accompanied the manifest file from The Clock Foundation, and is also available on the Clock Foundation GitHub. Of note: these cheetah annotations were from the 2018 reference genome Aci_jub_2 RefSeq (GCF_003709585.1) which is currently suppressed for genome annotation processing. We believe providing both the feline and cheetah gene annotations alongside the human annotations will enhance the interpretability of our results and support future homology-based analyses.

Data Availability: A link to the raw data is missing. For transparency and reproducibility, which is a key policy for PLOS ONE, please provide a link to a public repository where the data are available.

Our team is currently working to obtain the necessary permissions to upload the raw IDAT files, SESAME-normalized data received from The Clock Foundation, and the associated sample metadata sheet to NCBI GEO. We will provide the accession number once it becomes available.

Specific Comments

Abstract:

Please remove non-standard abbreviations from the abstract to adhere to journal guidelines.

Reworded so that differential methylation analysis (DMA) abbreviations were removed (Lines 28 and 30). Kept abbreviations for sinusoidal obstruction syndrome (SOS) to succinctly describe the case-control analysis done (Lines 30-31).

L24: I suggest changing the terminology from “over the age of maturation” to a more standard term like “adult” individuals, and specifying the age threshold used (e.g., age > 2 years).

Edited for clarity: “When applied to a test set of blood collected from live cheetahs, the clock provided accurate predictions for adult individuals (age > 3 years) but was less precise at and around age of sexual maturity” (Lines 24-25)

L30-33: The connection between these lines and the preceding sentences is unclear. Please revise to improve the flow of logic.

Reworded to connect the disease analysis to the age and sex analysis: “To explore the potential of methylation as a biomarker beyond age and sex, we conducted a differential methylation analysis to investigate disease-related methylation patterns in cheetahs diagnosed with hepatic sinusoidal obstruction syndrome (SOS). This analysis identified 4,377 CpG sites with significant differences between SOS-positive and SOS-negative cheetahs (adjusted p-value < 0.05). These findings advance the development of epigenetic clocks for precise age and sex prediction in cheetahs and related species and establish a foundation for leveraging methylation biomarkers to investigate diseases in wildlife conservation efforts.” (Lines 27-33)

Introduction:

L44: Please specify whether the epigenetic modifications mentioned correlate positively or negatively with age in this context.

Changed to indicate site-specific (can be up or down depending on site): “DNA methylation is one epigenetic modification that demonstrates site-specific linear changes that correlate with age” (Line 44)

L44-46: For clarity and conciseness, I suggest removing the phrase “and commonly occurs in regions of DNA where these cytosine residues are..”.

Removed the phrase “and commonly occurs in regions of DNA where these cytosine residues are”: “DNA methylation involves the covalent binding of methyl groups to cytosine residues. These regions are termed “CpG sites”, with the “p” denoting the phosphate linker between the two residues.” (Lines 44-46)

L47: Please edit "methyl groups" to the standard form "methyl-groups".

Changed “methyl groups” to “methyl-groups”: “The addition of methyl-groups results in structural changes to DNA that can affect gene expression” (Line 47)

L48: Please clarify what is meant by “structural changes” in this context.

Reworded and added an example of how methylation can structurally block transcription: “The addition of methyl-groups results in structural changes to DNA that can affect gene expression. For example, methylation at transcription factor binding sites can block transcription initiation[8].” (Lines 46-48)

L49: A reference is missing.

Added paper by Lu et al (reference #13) “Universal DNA methylation age across mammalian tissues” (Line 50)

L51: Please define what is meant by "Pan-species" upon first use.

Added definitions for both pan-species and pan-tissue for clarity: “Pan-species (i.e., applicable across multiple species) and pan-tissue (i.e., applicable across multiple tissue types) clocks such as the universal pan-mammalian epigenetic clocks developed by Lu et al[13] provide versatile tools for cross-species comparisons” (Lines 50-52)

L61-62: I suggest rephrasing “tools” to something more precise, such as “these traditional age-estimation methods are difficult to apply accurately”.

Reworded for clarity: “ Currently, skull measurements, radiographs, and dental examinations are the most common vertebrate age estimation methods[17–20]-- however, these methods are often difficult to apply accurately and consistently at scale in field settings.” (Lines 63-65)

L63: Please add "Measurements of" or "The study of" before "epigenetic aging" for logical clarity.

Reworded for clarity: “Epigenetic age clock measurements offer an alternative to these methods, with potential applications for determining not only chronological age but also health status.” (Lines 65-66)

L68: It is crucial information that all samples are from captive individuals with known chronological age, sex, and health status. Please state this clearly and early in the Introduction and Methods, as it is currently difficult for the reader to ascertain.

Reworded to clarify age and sex was known for all SDZWA and MCDB samples (used to build the clocks): “Building epigenetic clocks for wildlife is challenging due to limited biobanked samples, especially blood. At the San Diego Zoo Wildlife Alliance, we leveraged liver samples from our biobank and cheetah blood profiles from the Mammalian Consortium Data Browser (MCDB), all from captive individuals with known chronological age and sex, to construct an epigenetic clock applicable to less-invasive samples like blood and skin (compared to liver).” (Line 69)

Health status for MCDB samples not known but these samples were not used in health (SOS) analysis. Changed to specify health analysis done in SDZWA cheetahs: “Additionally, we used our extensive clinical and pathology database to investigate possible health status biomarkers in San Diego Zoo-housed cheetahs.” (Line 76)

The Introduction is missing an explicit mention of sex as a variable being investigated. Please incorporate this.

Currently, the last paragraph mentions both age and sex as variables investigated: “ 1) investigate DNA methylation patterns to develop cheetah-specific epigenetic clocks for age and sex prediction and expand this across clinically and field relevant tissues and wild felid species” (Line 80)

L75-81: This section is the ideal place to formulate the clear hypotheses or research questions mentioned in the Major Revisions.

Added clear hypotheses for the two specified goals: “We hypothesized that DNA methylation-based models could accurately predict age and sex in cheetahs, even with a limited sample size, and that incorporating multiple tissue types and related felid species would enhance predictive power and generalizability. Additionally, DNA methylation has been studied for its utility as a biomarker in various human and animal diseases [25,26], and we hypothesized that epigenetic differences would distinguish SOS-affected and unaffected cheetahs” (Lines 84-89)

Methods:

L83: It would be very helpful to state explicitly at the beginning of this section that "All animals were born and raised in captivity and held under human care until death," or similar phrasing.

Added for clarity: “The study population consisted of 447 cheetahs that had biobanked tissue samples available for use. All animals were born and held under human care until death.” (Line 96)

L89: Please change “die” to “died”.

Changed to “died”: “Samples were collected as part of a complete necropsy performed routinely on all animals that died at our facilities.” (Line 99)

L91: Please clarify if n=7 refers to the number of samples or individuals.

Changed to clarify that seven samples corresponding to seven individuals were used, and changed “blood” to “whole blood” to clarify sample type: “Biobanked whole blood samples (n = 7) collected from seven live cheetahs opportunistically during exams were also included” (Line 101).

L93-94: I suggest rephrasing; "depicts correlations" could be changed to "indicates repetitive sampling from the same individuals over time" if that is the intended meaning.

Rephrased for clarity on repetitive sampling from same individuals: “Table 1 summarizes the tissue types used in the study, and depicts repetitive sampling from the same individuals (Table 1)” (Lines 104-105)

L103: Since two different extraction kits were used, please confirm whether a potential batch effect was investigated and controlled for.

The clustering analysis performed on the samples didn’t show any distinct clustering based on the extraction kit used. We updated Supplemental Figure 1 to include trait bars that annotate sex, age, tissue, study and extraction kit to show this (image below), and edited the figure description in the manuscript to reflect these changes:

---

## [Decision Letter · Decision Letter 1]

5 Sep 2025

Dear Dr. Ysrael,

We look forward to receiving your revised manuscript.

Kind regards,

Nathan Wolf

Academic Editor

PLOS ONE

Journal Requirements:

Reviewers' comments:

Reviewer's Responses to Questions

**Comments to the Author**

Reviewer #1: All comments have been addressed

Reviewer #2: All comments have been addressed

2. Is the manuscript technically sound, and do the data support the conclusions?

Reviewer #1: Yes

Reviewer #2: Yes

3. Has the statistical analysis been performed appropriately and rigorously?

Reviewer #1: Yes

Reviewer #2: Yes

4. Have the authors made all data underlying the findings in their manuscript fully available?

Reviewer #1: No

Reviewer #2: No

5. Is the manuscript presented in an intelligible fashion and written in standard English?

Reviewer #1: Yes

Reviewer #2: Yes

Reviewer #1: The authors addressed most of my comments. Please provide a GSE number in the data availability section.

Reviewer #2: Dear authors,

thank you for addressing my comments.

Most of them have been solved sufficiently.

The onces missing concern the reproducibility of your results:

1) the data upload is not yet completed, as you wrote: "Our team is currently working to obtain the necessary permissions to upload the raw IDATfiles, SESAME-normalized data received from The Clock Foundation, and the associated

sample metadata sheet to NCBI GEO. We will provide the accession number once it becomes available." Please finalise before acceptance.

2) The method used for estimating the epigenetic clock has not been more described as I asked for, but less. Other authors will thus not been able to understand their coming about, nor to reprodcue the data. Please try to explain what has been done for your samples, at least by summarising the methods from the paper you mentioned.

**Do you want your identity to be public for this peer review?** For information about this choice, including consent withdrawal, please see our Privacy Policy

Reviewer #1: No

Reviewer #2: No

---

## [Author Response · Author response to Decision Letter 2]

16 Oct 2025

Dear Reviewers and Editors,

We thank you for your thoughtful and constructive feedback with the second round of comments. We appreciate your continued interest in this paper.

The two main concerns are 1) data availability and 2) methods from The Clock Foundation service we used:

Raw IDAT files, SESAME-normalized data from The Clock Foundation, and the sample metadata sheet are ready for submission to GEO. However, the GEO submission sheet requires fields to be filled out for the submission, detailing their conversion process and protocols used (labeling, hybridization, scanning protocols). We have contacted them for this information, and they let us know they are working on it, but they are taking a while. In the meantime, we have uploaded the IDATs, the IDAT metadata file with the protocol information we were able to fill out (extraction and normalization methods), and the normalized data file as a zipped folder (IDATS_normalized_data) for submission so the normalized data used in the study is available for use.

We have updated the manuscript to include more information on the Clock Foundation methods using information available to us. Please see below for more details.

In the manuscript text we incorporated reviewer-recommended references and updated supplementary material titles and descriptions. The renamed supplementary files were re-uploaded to match PLOS ONE style requirements. All edits are described in detailed in red text with line numbers referencing the revised manuscript.

Thank you again for your time and advice.

Sincerely,

Michelle Ysrael, Steven Kubiski, and Caroline Moore

---

## [Editor Report · Decision Letter 2]

21 Oct 2025

Investigating epigenetic biomarkers of age, sex, and disease in captive South African cheetahs (Acinonyx jubatus jubatus)

PONE-D-25-07460R2

Dear Dr. Ysrael,

We’re pleased to inform you that your manuscript has been judged scientifically suitable for publication and will be formally accepted for publication once it meets all outstanding technical requirements.

Kind regards,

Nathan Wolf

Academic Editor

PLOS ONE
---

## [Editor Report · Acceptance letter]

PONE-D-25-07460R2

PLOS One

Dear Dr. Ysrael,

I'm pleased to inform you that your manuscript has been deemed suitable for publication in PLOS One. Congratulations! Your manuscript is now being handed over to our production team.

Kind regards,

on behalf of

Dr. Nathan Wolf

Academic Editor

PLOS One